# Family Support and Social Support Associated with National Essential Public Health Services Utilization among Older Migrants in China: A Gender Perspective

**DOI:** 10.3390/ijerph19031610

**Published:** 2022-01-30

**Authors:** Wangnan Cao, Qingping Yun, Chun Chang, Ying Ji

**Affiliations:** Department of Social Medicine and Health Education, School of Public Health, Peking University, Beijing 100191, China; wangnancao@bjmu.edu.cn (W.C.); 1711110152@bjmu.edu.cn (Q.Y.); changchun@bjmu.edu.cn (C.C.)

**Keywords:** older migrants, National Essential Public Health Service, China, social support, family support, service utilization

## Abstract

China provides National Essential Public Health Services (NEPHS) free of charge to all citizens to ensure access to essential health services. The present study aimed to explore the associations between different sources of support and NEPHS service utilization among older migrants in China with a gender perspective. We used a national cross-sectional dataset derived from the 2015 China Migrants Dynamic Survey. Participants were included if they were aged ≥60 years and without household registration at the residence. Among 1989 participants, 35.2% enrolled in a free physical examination in the past year: 34.6% for males and 35.9% for females. Among male participants, having more local friends (OR = 1.47, 95% CI: 1.09, 1.99) and having insurance at the residence (OR = 1.75, 95% CI: 1.03, 2.96) were associated with enrolment in a free physical examination after controlling for age, education, and self-reported health status. Two additional factors, marital status and family structure, were found for female participants to be associated with enrolment in a free physical examination. NEPHS service utilization was far from satisfactory among older migrants in China, and both family support and social support played a role in it. There are common and unique factors associated with NEPHS service utilization in terms of gender.

## 1. Introduction

Health for all has become one of the critical efforts to achieve sustainable development goals (SDGs) worldwide and in China [1]. China provides National Essential Public Health Services (NEPHS) free of charge since 2009 to all citizens to ensure access to essential services (e.g., medical check-ups and chronic disease follow-up), regardless of gender, age, income, and place of residence. Compared to the general local population, older migrants are a vulnerable subpopulation reported with poor NEPHS utilization in the existing literature [2].

China has a large and increasing number of older migrants given population aging and rapid economic and social development. The number of older migrants aged over 60 reached 9.34 million, accounting for 7.2% of the total migrants [2]. Poor social interaction is commonly observed among older migrants [3,4], probably due to limited accessibility to local resources, difficulties in adapting to the local cultures, and limited capabilities to sustain the original network or develop new support networks [5,6]. To address these needs of migrants, China launched Essential Public Service Equalization for Migrants in 2013. Understanding the determinants of service utilization among older migrants becomes pressing in order to guide resource allocation, program design, and implementation.

We identified five publications in the existing literature that explored the prevalence and potential factors associated with NEPHS utilization among older migrants in China [7,8,9,10,11]. In general, these studies found that NEPHS utilization was unsatisfactory, e.g., 58% not establishing a health record, 64% not enrolling in a yearly medical check-up, and 46% not seeking medical service when needed. Identified barriers to NEPHS utilization included limited-service accessibility [12], poor health status, a limited number of local friends, and lacking health insurance [7,8,9,10,11]. Some of these identified barriers fell into the domain of support, which is an enabling factor of service utilization [13,14]. However, these studies did not differentiate different sources of support, e.g., family support and broader social support. Another limitation in the existing literature is lacking a particular gender perspective. We believe that gender difference should be prominent in terms of service utilization among older migrants, given the gender role differences in the family and different gender expectations in Chinese society.

The present study aimed to explore the associations between different sources of support and NEPHS utilization among older migrants in China from a gender perspective. Support was categorized into family support and social support in the present study, and we separated analyses for male and female migrants.

## 2. Materials and Methods

### 2.1. Data Sources

The data used in the present study were from a national cross-sectional dataset derived from the 2015 China Migrants Dynamic Survey (CMDS). CMDS is the largest survey in China among migrants, and the surveyed sample is highly representative due to its rigorous sampling strategy (a stratified, multi-stage, scale-oriented probability proportionate to the size method) and validated data collection process. More details can be found in the handbook [15], and its methods have been validated and reported in several previous studies [16,17,18,19]. Consistent with the NEPHS guideline, older migrants in the present study are defined as older people aged 60 years or older whose household registration is not in the local area but who have been living in the local area longer than a month. Two key NEPHS utilization indicators were used, including enrolment in a free physical examination for all migrants and enrolment in a chronic disease follow-up for migrants who have physician-diagnosed chronic diseases. Chronic diseases here are limited to hypertension and diabetes, per NEPHS guidelines.

### 2.2. Extraction of the Study Population

We extracted households with older adults aged 60 years or older from the 206,000 households in the 2015 CMDS data. If two older people were registered in the same household, random numbers were used to select one for inclusion in the study. In total, 9242 samples were obtained. Given the population risk profile and available resources, different age cut-offs (65 years cut-off to enroll in a free physical examination while 60 years cut-off to enroll in a chronic disease follow-up) were used per NEPHS service guideline. To analyze the enrolment in free physical examinations, we included 1989 individuals aged 65 years or older who had lived locally for more than six months. To analyze the enrolment in the chronic disease follow-up, we included 899 individuals aged 60 years or older who had lived locally for more than six months and had physician-diagnosed hypertension and/or diabetes mellitus.

### 2.3. Measurement of Variables

Demographics and health status: Gender, age, highest education obtained, and location of residence (Eastern, Central, Western, and Northeast per Administrative Divisions of the People’s Republic of China). The respondents were asked to self-rate their health status (healthy, unhealthy but able to take care of oneself, and unhealthy and unable to take care of oneself) and reported if they had a confirmed diagnosis for hypertension and/or diabetes by a physician.

NEPHS utilization: Two questions were asked: “Have you enrolled in a free physical examination provided by a community health center in the past year?” and “Have you enrolled in any follow-up services due to your hypertension or diabetes status in the past year?” Follow-up services included outpatient clinic visits, telephone follow-ups, or home visits. Responses included yes or no.

Family and social support: Two sources of support were asked, including family support and social support [20,21]. Information on family support included average monthly household income (obtained by dividing the total average monthly household income by the number of household members: CNY <1000, 1000–2000, >2000), marital status (widowed/single, married), and family structure (live alone, live with a spouse, live with children or grandchildren, and live with children and grandchildren). Information on social support included the number of local friends ((≥4 vs. <4) and health insurance status (no insurance, insurance at residence, and insurance in their hometown).

### 2.4. Statistical Analysis

Percentages were used to describe the basic information, and chi-square tests were performed for the one-way analysis. Logistic regression was conducted to explore the relationship between each support variable and NEPHS utilization. The odds ratio (OR) and 95% confidence intervals (CI) were used to describe the relationship between variables after controlling for background variables including age, education, health status, and location of residence. We did separate analyses for male and female participants. A two-sided test was performed set at α = 0.05. The study used SPSS 22.0 Windows (SPSS, Inc., Chicago, IL, USA) to analyze the data.

## 3. Results

### 3.1. Characteristics of Participants Eligible for Free Enrolment in a Free Physical Examination (Aged 65 Years and Older)

Among a total of 1989 participants who were eligible for enrolment in a free physical examination, half (51.9%) were males (*N* = 1032) and half (48.1%) were females (*N* = 957). Approximately half were aged between 65 and 70 years old, and the other half were older than 70 years. More than half had an education level below the primary school. Relatively more older people moved to the West (46.2%) and East (30.9%), and less to the Northeast (14.6%) and Central (8.4%). The majority (83.4%) of the participants rated themselves as healthy and 28.6% of the older migrants reported that they had hypertension and/or diabetes diagnosed by a physician. There was no gender difference in terms of these background characteristics. Of the 1989 survey respondents, 35.2% enrolled in a free physical examination in the past year, 34.6% for males and 35.9% for females (*p* > 0.05 between males and females) (Table 1).

The majority (70.0%) of the participants had health insurance in their hometown, 13.6% had health insurance at their residence, and 16.3% have neither. Two-thirds of the participants were married, 55.3% had less than three family members living together, and the average monthly household income was evenly distributed (CNY 35.3% < 1000, 34.5% 1000–2000, 30.1% ≥ 2000). Two-thirds of the participants had more than 4 local friends, 23.8% had 1–3 friends, and 9.1% had no local friends.

Among male participants, the number of local friends and health insurance status were associated with the enrolment in a free physical examination in the past year after controlling for age, education, and self-reported health status. Specifically, compared to participants with 0–3 local friends, participants with four or more local friends were more likely to be enrolled in a free physical examination in the past year (OR = 1.47, 95% CI: 1.09, 1.99). Compared to participants with no insurance, participants with health insurance at their residence were more likely to be enrolled in a free physical examination (OR = 1.75, 95% CI: 1.03, 2.96), and participants with health insurance in their hometown were also more likely to be enrolled in a free physical examination (OR = 1.83, 95% CI: 1.21, 2.77). Household income, marital status, and family structure were not associated with enrolment in a free physical examination (Table 2).

Among female participants, marital status, family structure, number of local friends, and health insurance status were associated with the enrolment in a free physical examination in the past year after controlling for age, education, and self-reported health status. Specifically, compared to widowed/single participants, married participants were more likely to be enrolled in a free physical examination in the past year (OR = 1.55, 95% CI: 1.10, 2.20). Compared to participants living alone, participants living with children and grandchildren were less likely to be enrolled in a free physical examination in the past year (OR = 0.50, 95% CI: 0.28, 0.90); no difference was found between participants living with a spouse. Compared to participants with 0–3 local friends, participants with four or more local friends were more likely to be enrolled in a free physical examination in the past year (OR = 1.63, 95% CI: 1.21, 2.20). Compared to participants with no insurance, participants with health insurance at their residence were more likely to be enrolled in a free physical examination (OR = 1.71, 95% CI: 1.05, 2.79); no difference was found for participants with health insurance at their hometown. Household income was not associated with enrolment in a free physical examination (Table 2).

### 3.2. Characteristics of Participants with Chronic Disease Eligible for Free Enrolment in a Follow-Up Service (Aged 60 Years and Older)

Among a total of 899 older migrant participants with chronic disease eligible for free enrolment in a follow-up service, around half (48.8%) of the older migrant population were males (*N* = 439) and half (51.2%) were females (*N* = 460). Two-thirds (65.9%) were aged less than 70 years. The majority of the participants had an education level below primary school. More participants live in the Western (44.5%) and Eastern regions (34.9%), and fewer in the Northeast (13.9%) and Central (6.7%). The majority (74.1%) of survey respondents rated themselves as healthy. Among the 899 survey respondents with a chronic disease, 34.3% enrolled in a follow-up service in the past year, 34.6% for males and 33.9% for females (*p* > 0.05 between males and females) (Table 3).

The majority (73.3%) of the participants had health insurance in their hometown, 14.6% had health insurance at their residence, and 12.1% had neither. More than two-thirds (72.0%) of the participants were married, 50.9% had less than three family members living together, and the per capita monthly household income was evenly distributed (CNY 32.0% < 1000, 36.6% 1000–2000, 31.4% ≥ 2000). Of the survey respondents, 66.1% had four and more local friends, 24.7% had 1–3 friends, and 9.2% had no local friends. The median number of local friends was 5.

Among male participants, we did not find any support-related variables associated with the enrolment in a follow-up service for chronic disease in the past year after controlling for age, education, and self-reported health status (Table 4).

Among female participants, after controlling for age, education, and self-reported health status, participants with health insurance at their residence were more likely than those with no insurance to be enrolled in a follow-up service for chronic disease (OR = 2.47, 95% CI: 1.07, 5.69); no difference was found for participants with health insurance in their hometown. Other social-related variables were not associated with enrolment in a follow-up service for chronic disease (Table 4).

## 4. Discussion

The present study aimed to explore the associations between different sources of support and NEPHS utilization among older migrants in China from a gender perspective. We found a low NEPHS utilization among older migrants in China, and both family support and social support played a role in it. There are common (e.g., more local friends) and unique (e.g., family structure) support-related factors associated with NEPHS utilization in terms of gender. For example, having more local friends and having health insurance at the location of residence were associated with more NEPHS utilization for both genders. Married participants reported more NEPHS utilization than non-married participants, but this finding applied to females only.

We found low NEPHS utilization among older migrants in China, which was around half of the rates compared to the utilization among the local older population. The Fifth Health Services Survey in China in 2013 showed that 59.6% of the population aged 65 years or older had physical examinations in the past one year, while 71.3% of hypertensive patients aged 35 years or older had received follow-ups in the past three months [22]. This low utilization rate might be explained by imbalanced resources allocation across the country and limited service provisions at local facilities. We found that older migrants living in the Eastern region utilized NEPHS less than those living in other regions (27% vs. 39% for medical check-ups; 25% vs. 39% for disease follow-up). The Eastern region receives the most health resources and NEPHS funding across all regions, but it attracts a large number of migrants, particularly a young working-age migrant population, which might place high pressure on the NEPHS service provisions. As a result, there might be limited workforce or resources allocated to the older population. Therefore, older migrants were a particularly disadvantaged subpopulation in enrolling in NEPHS service utilization, which places them at a higher risk of late disease diagnosis, poor health status, and high disease burden.

In terms of gender, two common factors associated with NEPHS utilization were health insurance status and the number of local friends. Having health insurance at their residence allows them to access information easier and functions as a sign of good social integration and connectiveness to the local community [23]. Similarly, a large number of local friends also implies a high level of social integration and social support [24,25], and also good accessibility to local resources [26].

Low social support was found among this older migrant population due to low coverage of health insurance at their residence and a low number of local friends. The health insurance coverage rate (84%) was far below the rate among the counterpart local population, with a national average of 98.4% for local people aged 60 years and older [22]. One possible reason is the requirement of the health insurance enrollment fee (CNY 250–320 per year depending on the location), which migrants may have to pay both in their hometown (even if they do not live there) and their residence location in order to have comprehensive medical coverage. We also found that 10% of the older migrants had no local friends, with a median number of friends of five. This was far lower than the number (5–30) of friends reported by local older people, although the number was sometimes not directly comparable due to various samples of participants and different measurements on friends (e.g., closeness level) [27,28,29].

Two family support-related variables, marital status and family structure, were associated with NEPHS utilization, but only among female migrants. We found that married participants were more likely to be enrolled in a physical examination, which might be explained by the fact that older migrants were more dependent on family resources to utilize health care services [30,31,32] and combined resources from the couples. Unexpectedly, we found that an increased family size (living with children and grandchildren) was a barrier in NEPHS service utilization among female migrants. A possible reason could be that an increased family size means an increase in household activities such as home care particularly for female household heads and therefore limited time to utilize care. Few studies have investigated this family structure issue associated with service utilization, and further exploration is suggested [33,34].

We found household income was not associated with enrolment in a physical examination and chronic disease follow-up service, which was different from previous studies focusing on treatment-related service unitization [4]. Since NEPHS services were all free of charge, income should not be a barrier in accessing these services. However, there are other costs associated with service unitization, such as transportation costs to the local facility and time costs needed to spend on the service.

This study is subjected to some limitations. First, this was a cross-sectional study, and no causal inferences can be made based on these data. In the present study, we used association (rather than casual) terms to interpret our findings. Second, our measurements of family support and social support were self-constructed based on existing variable information and might not be comprehensive. To minimize potential bias, we took only objective measures related to support in the database and performed quality checks on the measures we included. Other key variables that reflected these supports (e.g., emotional support) might not have been captured in the survey. Third, we used self-reported data, and there might be recall bias given that the participants were older. However, the questionnaire was validated particularly among this population, and all interviewers were well-trained and experienced in collecting these data.

Despite these limitations, the present study explored the relationships between two sources of support (family support and social support) and NEPHS service utilization among the older migrant population. Both common and unique factors in terms of gender were explored to better aid resources allocation and programs design. CMDS is the largest and most representative survey targeted at old migrants, which makes the findings solid and valuable for policymaking.

## 5. Conclusions

Despite the fact that NEPHS services in China are free of charge to anyone, the NEPHS service utilization was far from satisfactory among older migrants—a highly vulnerable subpopulation needing support. For both genders, we found two social support indicators associated with NEPHS service utilization, including a large number of local friends and health insurance at the location of residence. We also found that married participants and participants living in a small household were more likely to use NEPHS services, but this finding applied to female migrants only. We conclude that a supportive community with easy access to health insurance and opportunities to know peers and build relationships is desirable for older migrants in China. Unmarried older migrants and migrants living in large households with children and grandchildren are particularly vulnerable and should be given extra attention. NEPHS is essential in ensuring health equity and achieving universal health coverage in China, but we acknowledge that sufficient NEPHS services’ utilization requires active participation by the individual (the user side) and also strong promotion from the government (the supply side).

## Figures and Tables

**Table 1 ijerph-19-01610-t001:** Participants’ characteristics in terms of enrolment in the free physical examinations among older migrants (*N* = 1989).

Variable Name	Category	Male				Female			
		Total Participants *N* (%)	Participation Enrolled in Free Physical Examination *n* (%)	x^2^	*p*	Total Participants *N* (%)	Participation Enrolled in Free Physical Examination *n* (%)	x^2^	*p*
Total sample	---	1032 (100.0)	357 (34.6)			957 (100.0)	344 (35.9)		
Age (years)	65–70	563 (54.6)	193 (34.3)	0.05	0.82	484 (50.6)	191 (39.5)	5.26	0.02 *
	70+	469 (45.4)	164 (35.0)			473 (49.4)	153 (32.3)		
Education level	Elementary school or below	595 (50.6)	193 (32.9)	2.89	0.09	700 (73.1)	234 (33.4)	7.17	0.007 **
	Junior school or above	437 (49.4)	164 (37.5)			257 (26.9)	110 (42.8)		
Health self-assessment	Healthy	879 (85.2)	313 (35.6)	3.09	0.21	779 (81.4)	282 (36.2)	8.37	0.04 *
	Unhealthy but able to take care of oneself	135 (13.1)	40 (29.6)			146 (15.3)	57 (39.0)		
	Unable to take care of oneself	18 (1.7)	4 (22.2)			32 (3.3)	5 (15.6)		
Monthly personal income per household (CNY)	<1000	353 (34.2)	114 (32.3)	1.52	0.47	350 (36.6)	128 (36.6)	0.92	0.63
	1000–2000	346 (33.5)	127 (36.7)			341 (35.6)	116 (34.0)		
	>2000	333 (32.3)	116 (34.8)			266 (27.8)	100 (37.6)		
Marital status	Widowed/single	225 (21.8)	66 (29.3)	3.52	0.06	435 (45.5)	134 (30.8)	9.16	0.02 *
	Married	807 (78.2)	291 (36.1)			522 (54.5)	210 (40.2)		
Family structure	Live alone	64 (6.2)	18 (28.1)	6.68	0.08	62 (6.5)	29 (46.8)	1.98	0.007 **
	Live with spouse	409 (39.6)	160 (39.1)			275 (28.7)	116 (42.2)		
	Live with children or grandchildren	184 (17.8)	61 (33.2)			158 (16.5)	53 (33.5)		
	Live with children and grandchildren	375 (36.3)	118 (31.5)			462 (48.3)	146 (31.6)		
Number of local friends	<4	303 (29.4)	84 (27.7)	8.95	0.003 **	362 (37.8)	100 (27.6)	17.51	<0.001 ***
	≥4	729 (70.6)	273 (37.4)			595 (62.2)	244 (41.0)		
Health insurance	No health insurance	153 (14.8)	35 (22.9)	10.90	0.004 **	172 (18.0)	60 (34.9)	16.31	<0.001 ***
	Health insurance at residence	139 (13.5)	51 (36.7)			132 (13.8)	68 (51.5)		
	Health insurance at hometown	740 (71.7)	271 (36.6)			653 (68.2)	216 (33.1)		
Residence location	Eastern	319 (30.9)	93 (29.2)	14.60	0.002 **	292 (30.5)	74 (25.3)	21.66	<0.001 ***
	Central	89 (8.6)	21 (23.6)			77 (8.0)	35 (45.5)		
	Western	468 (45.4)	185 (39.5)			446 (46.6)	181 (40.6)		
	Northeast	154 (14.9)	58 (37.7)			135 (14.1)	51 (37.8)		
Diagnosed hypertension or diabetes	Yes	269 (26.1)	106 (39.4)	3.72	0.05	299 (31.2)	114 (38.1)	0.90	0.34
	No	763 (73.9)	251 (32.9)			658 (68.8)	230 (35.0)		

* *p* < 0.05, ** *p* < 0.01, *** *p* < 0.001.

**Table 2 ijerph-19-01610-t002:** Multivariate logistic regression on factors associated with enrolment in free physical examination.

Variable Name	Category	Male		Female	
		OR	95% CI	OR	95% CI
Monthly personal income per household (CNY)	≤1000	1.00		1.00	
	1000–2000	1.31	0.95, 1.81	0.96	0.69, 1.34
	>2000	1.19	0.83, 1.70	1.15	0.78, 1.70
Marital status	Widowed/single	1.00			1.00
	Married	1.21	0.83, 1.75	1.55 *	1.10, 2.20
Family structure	Live alone	1.00		1.00	
	Live with spouse	1.37	0.72, 2.60	0.64	0.33, 1.25
	Live with children or grandchildren	1.05	0.54, 2.04	0.53	0.28, 1.01
	Live with children and grandchildren	1.08	0.58, 2.01	0.50 *	0.28, 0.90
Number of local friends	<4	1.00		1.00	
	≥4	1.47 *	1.09, 1.99	1.63 **	1.21, 2.20
Health insurance	No health insurance	1.00			1.00
	Health insurance at residence	1.75 *	1.03, 2.96	1.71 *	1.05, 2.79
	Health insurance at hometown	1.83 **	1.21, 2.77	0.83	0.57, 1.21
Residence location	Eastern	1.00		1.00	
	Central	0.72	0.41, 1.27	2.43 **	1.38, 4.25
	Western	1.68 **	1.20, 2.34	2.10 **	1.46, 3.01
	Northeast	1.53	0.98, 2.39	1.52	0.94, 2.45

The models were controlled for age, education, and health self-assessment. * *p* < 0.05, ** *p* < 0.01.

**Table 3 ijerph-19-01610-t003:** Participants’ characteristics in terms of enrolment in the chronic disease follow-up among older migrants (*N* = 899).

		Male				Female			
Variable Name	Category	Total Participants *N* (%)	Participation Enrolled in Chronic Disease Follow-Up *n* (%)	x^2^	*p*	Total Participants *N* (%)	Participation Enrolled in Chronic Disease Follow-Up *n* (%)	x^2^	*p*
Total sample	---	439 (100.0)	152 (34.6)			460 (100.0)	156 (33.9)		
Age (years)	65–70	292 (66.5)	96 (32.9)	1.17	0.27	300 (65.2)	91 (30.3)	4.93	0.03 *
	70+	147 (33.5)	56 (38.1)			160 (34.8)	65 (40.6)		
Education level	Elementary school or below	212 (48.3)	70 (33.0)	0.47	0.50	317 (68.9)	108 (34.6)	0.01	0.92
	Junior school or above	227 (51.7)	82 (36.1)			143 (31.1)	48 (33.6)		
Health self-assessment	Healthy	328 (74.7)	110 (33.5)	1.25	0.54	338 (73.5)	109 (32.2)	1.80	0.41
	Unhealthy but able to take care of oneself	100 (22.8)	39 (39.0)			106 (23.0)	40 (37.7)		
	Unable to take care of oneself	11 (2.5)	3 (27.3)			16 (3.5)	7 (43.8)		
Monthly personal income per household (CNY)	<1000	134 (30.5)	40 (29.9)	3.33	0.19	154 (33.5)	51 (33.1)	2.44	0.30
	1000–2000	161 (36.7)	64 (39.8)			168 (36.5)	64 (38.1)		
	>2000	144 (32.8)	48 (33.3)			138 (30.0)	41 (29.7)		
Marital status	Widowed/single	68 (15.5)	23 (33.8)	0.02	0.88	184 (40.0)	68 (37.0)	1.27	0.26
	Married	371 (84.5)	129 (34.8)			276 (60.0)	88 (31.9)		
Family structure	Live alone	21 (4.8)	6 (28.6)	4.15	0.25	23 (5.0)	9 (39.1)	0.73	0.87
	Live with spouse	163 (37.1)	60 (36.8)			126 (27.4)	42 (33.3)		
	Live with children or grandchildren	65 (14.8)	28 (43.1)			76 (16.5)	28 (26.8)		
	Live with children and grandchildren	190 (43.3)	58 (30.5)			235 (51.1)	77 (32.8)		
Number of local friends	<4	136 (31.0)	41 (30.1)	1.75	0.19	169 (36.7)	54 (32.0)	0.46	0.50
	≥4	303 (69.0)	111 (36.1)			291 (63.3)	102 (35.1)		
Health insurance	No health insurance	50 (11.4)	17 (34.0)	0.57	0.75	59 (12.8)	15 (25.4)	7.87	0.02 *
	Health insurance at residence	59 (13.4)	23 (39.0)			72 (15.7)	34 (47.2)		
	Health insurance at hometown	330 (75.2)	112 (33.9)			329 (71.5)	107 (32.5)		
Residence location	Eastern	156 (35.5)	40 (25.6)	9.80	0.02 *	155 (33.7)	37 (23.9)	23.55	<0.001 **
	Central	28 (6.4)	13 (46.4)			32 (7.0)	18 (56.3)		
	Western	191 (43.5)	72 (37.7)			206 (44.8)	86 (41.7)		
	Northeast	61 (13.9)	26 (42.6)			63 (13.7)	14 (22.2)		

* *p* < 0.05, ** *p* < 0.01.

**Table 4 ijerph-19-01610-t004:** Multivariate logistic regression on factors associated with enrolment in chronic disease follow-up.

Variable Name	Category	Male		Female	
		OR	95% CI	OR	95% CI
Monthly personal income per household (CNY)	≤1000	1.00		1.00	
	1000–2000	1.60	0.96, 2.67	1.35	0.83, 2.19
	>2000	1.63	0.90, 2.93	1.10	0.62, 1.95
Marital status	Widowed/single				
	Married	1.05	0.56, 1.97	0.83	0.51, 1.36
Family structure	Live alone	1.00		1.00	
	Live with spouse	1.53	0.49, 4.75	0.78	0.27, 2.25
	Live with children or grandchildren	2.10	0.65, 6.78	0.75	0.26, 2.15
	Live with children and grandchildren	1.35	0.45, 4.05	0.63	0.24, 1.65
Number of local friends	<4				
	≥4	1.29	0.82,2.03	1.17	0.76,1.79
Health insurance	No health insurance	1.00		1.00	
	Health insurance at residence	0.99	0.43, 2.28	2.47 *	1.07, 5.69
	Health insurance at hometown	0.85	0.44, 1.66	1.25	0.63, 2.46
Residence location	Eastern	1.00		1.00	
	Central	2.91 *	1.23, 6.87	3.33 **	1.44, 7.70
	Western	2.05 **	1.22, 3.45	2.13 **	1.29, 3.52
	Northeast	2.59 **	1.29, 5.21	0.72	0.33, 1.56

The models were controlled for age, education, and health self-assessment. * *p* < 0.05, ** *p* < 0.01.

## Data Availability

Restrictions apply to the availability of these data. Data were obtained from the Migrants Population Service Center, National Health Commission P.R. China and are available from the authors with the permission of the National Health Commission P.R. China.

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
