# Peer review of "Family Support and Social Support Associated with National Essential Public Health Services Utilization among Older Migrants in China: A Gender Perspective"

_ijerph, 2022, doi:10.3390/ijerph19031610_

Round 1
Reviewer 1 Report
This study analyzed the associations between different sources of support and NEPHS service utilization among older migrants in China with a gender perspective.
Although the topic of this study is interesting, it has some serious limitations. First, Overall, this study is concise and lacks rich discussion. Second, the introduction does not contain sufficient background and include all relevant references. Third, this study mainly analyzes descriptive statistics and associations between variables, therefore it is questionable what theoretical and practical implications these analysis results have. The author(s) needs to persuasively explain to the reader what kind of differentiation this study has compared to previous studies, but this study does not present any relevant contents at all. In my opinion, this is more of a report rather than an academic paper.
Author Response
Response: Thanks for taking the time to review our manuscript and to provide us constructive comments. In this revised manuscript, we have added relevant references, provided more details on the study design, elaborated on our findings in the Discussion, and better interpreted our results given the study limitation. Details can be found in the revised manuscript (with track changes). We hope you find the manuscript improved. Thanks.
Reviewer 2 Report
Family support and social support associated with National Essential Public Health Services (NEPHS) utilization among older migrants in China: A gender perspective
This is an important manuscript that adds to the literature on factors affecting utilization of free Public health care particularly in China. The manuscript is well written, and it used a representative sample. However, the following minor comments can improve the manuscript.
Abstract
- Page 1, line 15. The authors stated “34.6% for males and 35.9% for females”. This does not add up to 100%.
- Page 1, line 15-17: “Among male participants, the number of local friends and health insurance status were associated with enrolment in a free physical examination after controlling for age, education, and self-reported health status”. It is good to add the p-value or OR .
Methods
- Page2, line 83-85: “Different age cut-offs (65 years cut-off to enrol in a free physical examination while 60 years cut-off to enrol in a chronic disease follow-up) were used per NEPHS service guideline”. Though it can be found in the guideline, it is good to briefly explain this in the manuscript.
Results
- Page 3, line 128: As mentioned above, “34.6% for males and 35.9% for females” does not add up to 100%.
Discussion
- In the beginning of the discussion, it is good to restate the objective of the study
- Page 7, line 201-206: This looks like a summary of findings of the study and it is not clear to readers.
- Page 7, line 231-23s: “The health insurance coverage rate (84%) was far below the rate among the counterpart local population, with a national average of 98.4% for local people aged 60 years and older”. Authors should provide possible reason for the low health insurance enrolment. Is the enrolment free of charge or at a fee? If at a fee, how much is the enrolment fee.
- Page 7, line 239-240: “We found that married participants were more likely to be enrolled in a physical examination…”. This could be due to combined resources from the couples.
- Page 7, line 242-244: “Unexpectedly, we found that increased family size (living with children and grandchildren) was a barrier in NEPHS service utilization among female migrants”. A possible reason could be that increased family size means increase in household activities such as home care particular for female household heads and therefore limited time to utilize care.
- Page 8, line 248-249: “Since NEPHS services were all free of charge, income should not be a barrier in accessing these services”. Authors should elaborate on this for readers to know whether there is no distance to seek the care given that transportation cost could be a hindrance.
- Page 8, line 250-261: It is not enough to just mention that limitations of the study. It is good to mentioned how you were able to minimize the limitations in order not affect the study findings. Or to explain why the method you used has a minimum effect of the findings.
References
Reference 13 is very old.
Author Response
Reviewer #2:
This is an important manuscript that adds to the literature on factors affecting utilization of free Public health care particularly in China. The manuscript is well written, and it used a representative sample. However, the following minor comments can improve the manuscript.
Response: We appreciate your positive feedback and constructive comments below.
Abstract
- Page 1, line 15. The authors stated “34.6% for males and 35.9% for females”. This does not add up to 100%.
Response: Here, we report the separate data on males and females regarding the prevalence of enrolment in a free physical examination in the past year (34.6% for males and 35.9% for females). These two numbers do not need to be added up to 100%.
- Page 1, line 15-17: “Among male participants, the number of local friends and health insurance status were associated with enrolment in a free physical examination after controlling for age, education, and self-reported health status”. It is good to add the p-value or OR .
Response: We have added OR numbers accordingly. Thanks.
Methods
- Page2, line 83-85: “Different age cut-offs (65 years cut-off to enrol in a free physical examination while 60 years cut-off to enrol in a chronic disease follow-up) were used per NEPHS service guideline”. Though it can be found in the guideline, it is good to briefly explain this in the manuscript.
Response: Thanks. We have elaborated a bit more on this:
“Given the population risk profile and available resources, different age cut-offs (65 years cut-off to enrol in a free physical examination while 60 years cut-off to enrol in a chronic disease follow-up) were used per NEPHS service guideline.”
Results
- Page 3, line 128: As mentioned above, “34.6% for males and 35.9% for females” does not add up to 100%.
Response: (Same as above)
Discussion
- In the beginning of the discussion, it is good to restate the objective of the study
Response: Good point. We have re-stated the study objective as below.
“The present study aimed to explore the associations between different sources of support and NEPHS utilization among older migrants in China with a gender perspective.”
- Page 7, line 201-206: This looks like a summary of findings of the study and it is not clear to readers.
Response: We have revised this section as below to improve its clarify.
“We found a low NEPHS utilization among older migrants in China, and both family support and social support played a role in it. There are common (e.g., more local friends) and unique (e.g., family structure) support-related factors associated with NEPHS utilization in terms of gender. For example, having more local friends and having health insurance at the location of residence were associated with more NEPHS utilization for both genders. Married participants reported more NEPHS utilization than non-married participants, but this finding applied to females only.”
- Page 7, line 231-23s: “The health insurance coverage rate (84%) was far below the rate among the counterpart local population, with a national average of 98.4% for local people aged 60 years and older”. Authors should provide possible reason for the low health insurance enrolment. Is the enrolment free of charge or at a fee? If at a fee, how much is the enrolment fee.
Response: Health insurance enrolment is at a fee (CNY 250-320 per year depending on location). We have added this information to the revised manuscript.
- Page 7, line 239-240: “We found that married participants were more likely to be enrolled in a physical examination…”. This could be due to combined resources from the couples.
Response: True! We have added this to the revised sentence.
- Page 7, line 242-244: “Unexpectedly, we found that increased family size (living with children and grandchildren) was a barrier in NEPHS service utilization among female migrants”. A possible reason could be that increased family size means increase in household activities such as home care particular for female household heads and therefore limited time to utilize care.
Response: We greatly appreciate this insight and have added this explanation to the revised manuscript.
- Page 8, line 248-249: “Since NEPHS services were all free of charge, income should not be a barrier in accessing these services”. Authors should elaborate on this for readers to know whether there is no distance to seek the care given that transportation cost could be a hindrance.
Response: Thanks. We have elaborated this as below:
“Since NEPHS services were all free of charge, income should not be a barrier in accessing these services. However, there are other costs associated with service unitization, such as transportation cost to the local facility and time cost needed to spend on the service.”
- Page 8, line 250-261: It is not enough to just mention that limitations of the study. It is good to mentioned how you were able to minimize the limitations in order not affect the study findings. Or to explain why the method you used has a minimum effect of the findings.
Response: We have elaborated on what we did to minimize potential bias on this study.
“This study is subjected to some limitations. First, this was a cross-sectional study, and no causal inferences can be made based on these data. In the present study, we used association (rather than casual) terms to interpret our findings. Second, our measurements on family support and social support were self-constructed based on existing variable information and might not be comprehensive. To minimize potential bias, we took only objective measures related to support in the database and did quality checks on the measures we included. Other key variables that reflected these support (e.g., emotional support) might be failed to be captured in the survey. Third, we used self-reported data, and there might be recall bias given that the participants were older. However, the questionnaire was validated particularly among this population, and all interviewers were well-trained and experienced in collecting these data.
References
Reference 13 is very old.
Response: We have replaced this old reference with a new one below:
“13. Babitsch B, Gohl D, Lengerke T: Re-revisiting Andersen's Behavioral Model of Health Services Use: a systematic review of studies from 1998-2011. Psychosoc Med 2012, 9 (11).”
Round 2
Reviewer 1 Report
The authors revised mainly discussion. Through this, they supplemented the interpretation of the analysis results in this study. The clarity of expression and readability, such as sentence structure, jargon use, acronyms, etc. needs to be supplemented.
Author Response
Thanks for the positive feedback and constructive comments. We have edited the manuscript one more time throughout. Thanks Again!